# Exogenous Selenium Enhances Cadmium Stress Tolerance by Improving Physiological Characteristics of Cabbage (*Brassica oleracea* L. var. *capitata*) Seedlings

**Kaiyue Jia, Zhipeng Zhan, Bingqian Wang, Wuhong Wang, Wenjing Wei, Dawei Li, Wei Huang \* and Zhongmin Xu \***

College of Horticulture, Northwest A&F University, Xianyang 712100, China; kaiyuejia1998@nwafu.edu.cn (K.J.); 2022055322@nwafu.edu.cn (Z.Z.); wbq2021@nwafu.edu.cn (B.W.); wwh6169@nwafu.edu.cn (W.W.); wwjing81@nwafu.edu.cn (W.W.); xndavid@nwsuaf.edu.cn (D.L.)
\* Correspondence: xnhw@nwsuaf.edu.cn (W.H.); xnxzm@nwafu.edu.cn (Z.X.)

**Abstract:** In recent years, the levels of cadmium (Cd) in agricultural soils have been increasing. Cd is highly toxic and can enter the human body through the food chain, threatening human health, therefore, reducing the Cd content in vegetables and producing green and non-polluting food has become a common concern in society. However, the physiological properties of exogenous selenium in alleviating Cd stress in cabbage seedlings have not been thoroughly investigated. In this study, exogenous Se (10 μMol/L) was applied under Cd (25 μMol/L) stress and the physiological characteristics such as biomass, photosynthetic fluorescence parameters, Se and Cd contents, chloroplast ultrastructure, leaf membrane esterification, and antioxidant enzyme activities were determined. The results showed that the exogenous application of Se could effectively alleviate the decrease in growth, photosynthetic pigment, and the gas exchange characteristics of the cabbage seedlings under Cd stress, improve cabbage root vitality, reduce root leaf Cd content, and alleviate the Cd stress-induced damage. Ultrastructural observation showed that the Cd stress caused the disruption to the chloroplasts' internal structure in the cabbage leaves, while an exogenous Se treatment alleviated the chloroplast damage to some extent, improved the stability of the inner capsule membrane, and alleviated the Cd stress-induced damage to the photosynthetic organs. Cd stress also caused oxidative damage and the excessive accumulation of reactive oxygen species (ROS) in the leaves of cabbage seedlings, as evidenced by the significant accumulation of superoxide anion ($O^{2-}$), hydrogen peroxide ($H_2O_2$), malondialdehyde (MDA), and electrolyte leakage. On the other hand, after the exogenous Se treatment, the Cd stress-induced oxidative damage could be reduced by up-regulating the activities of the antioxidant enzymes, such as superoxide dismutase (SOD), peroxidase (POD), and ascorbate peroxidase (APX). At the same time, Cd stress significantly increased glutathione (GSH) levels, and the exogenous Se treatment further increased the GSH levels, thereby increasing the tolerance of the cabbage to Cd stress. In conclusion, exogenous Se can further improve the Cd tolerance of the cabbage seedlings by protecting the photosynthetic system, eliminating excessive accumulation of reactive oxygen species under Cd stress, alleviating oxidative stress, and reducing Cd levels in plants, among other physiological properties.

**Keywords:** cabbage (*Brassica oleracea* L. var. *capitata*); cadmium stress; photosynthesis; physiological characteristics; selenium

## 1. Introduction

With the rapid development of industry, heavy metal pollution caused by mining, waste incineration, and the heavy use of pesticides and fertilizers has become increasingly serious [1]. Cd is a type of harmful and non-essential toxic heavy metal for plants, and once absorbed, it inhibits the plant growth and development, resulting in slower root development, curled or chlorotic and etiolated leaves, and in severe cases, plant death [2,3].

Cd inhibits plant growth by affecting physiological processes such as photosynthesis, oxidative damage, and mineral nutrition, resulting in growth retardation, leaf yellowing, and biomass reduction. Some studies have shown that the chloroplast granule and thylakoid lamellar structure of wheat were disrupted under Cd stress, and the efficiency of light capture and photosynthetic electron transport were significantly reduced [4,5]. Many studies have reported that active oxygen species and membrane peroxidation of rapeseed and *Arabidopsis thaliana* increased under Cd stress [6,7]. In addition, Cd not only affects plant growth and development, but also has a negative impact on human health when it accumulates in plants and enters the food chain. Therefore, agricultural, and environmental researchers have focused on finding a method that can effectively mitigate Cd stress on crops.

In recent years, the exogenous application of plant growth regulators has become an important way of reducing Cd damage to plants and even reducing Cd uptake. It has been reported that external foliar application of fulvic acid can reduce Cd toxicity in lettuce [8] and that ascorbic acid can reduce Cd toxicity and absorption in maize [9]. During plant growth, tolerance to abiotic stress increased after Se application [10,11]. Studies have shown that Se can counteract and reverse the harmful effects of Cd on plants, improve plant photosynthesis, regulate the balance of plant mineral elements, reduce Cd levels in plants, and increase plant antioxidant capacity to alleviate Cd stress in plants. FILEK et al. found that Cd stress caused chloroplast membrane degradation in rapeseed, whereas the exogenous Se application resulted in the ultrastructural reconstruction of the chloroplast, recombination of the thylakoid and stromal lamellar structure, and increased fatty acid unsaturation of chloroplast size and fluidity of the plasma membrane [12]. Zhang et al. found that the surface application of Se increased photosystem II and the electron transfer rate of *Brassica napus* leaves under Cd stress, while the root application of Se increased the photosynthetic rate, stomatal conductance, transpiration rate, and stomatal threshold under Cd stress [13]. Se can enhance the capacity of the plant antioxidant defense system, reduce membrane lipid peroxidation, and increase resistance to oxidative stress by scavenging intracellular free radicals, increasing antioxidant oxidase activity and non-enzymatic antioxidant content [14].

Cabbage (*Brassica oleracea* L. var. *capitata*) occupies a very important position in the national annual vegetable supply and is exported. As a cruciferous plant, cabbage usually has a strong ability to absorb and accumulate heavy metals [15], and the increased Cd content in the soil and farmland led to a significant reduction in the yield and quality. Therefore, it is a matter of concern to reduce the accumulation of Cd in cabbage. This study aimed to investigate the protective mechanism of exogenous Se on Cd stress in cabbage seedlings, that is, that exogenous Se may improve the growth, photosynthesis, and gas exchange ability of cabbage to protect the chloroplast ultrastructure of leaves, prevent oxidative damage by enhancing antioxidant activity, and mediate the absorption and transport of Cd to enhance the tolerance of plants to Cd. This study provides a reference for exploring the physiological properties related to the alleviation of Cd stress in kale seedlings by exogenous Se.

## 2. Materials and Methods

### 2.1. Plant Materials and Handling

The cabbage variety 'Fuer' was provided by the College of Horticulture, Northwest A&F University. Seeds were soaked in 5% sodium hypochlorite for 5 min, rinsed twice with distilled water, and then germinated for 24 h in the dark at 25 °C.

Seeds were planted in a hollow dish with a soil: vermiculite ratio of 3:1 and grown in an incubator at 25/20 °C (day/night), with a 14 h/10 h (day/night) photoperiod, 150 $\mu$mol·m$^{-2}$·s$^{-1}$ light intensity, and 70–80% relative humidity. Seedlings were transplanted into a plastic pot (25 cm × 20 cm × 10 cm; six plants per pot). To ensure normal growth of the cabbage, the Cd and Se concentration tests were carried out after 2 days

of slow germination in the Hogland's nutrient solution, and the nutrient solution was renewed every 5 days.

The experiment was a completely randomized block design with four treatments: CK: Hoagland's solution; Se: Hoagland's solution + 10 mg/L Se; Cd: Hoagland's solution + 25 μMol/L Cd; Cd + Se: Hoagland's solution + 25 μMol/L Cd + 10 mg/L Se, where the concentration of Cd (25 μMol/L) was chosen according to the previous study [16,17], Se by exogenous spraying on leaves of the kale seedlings [18]. It was grown at 25/20 °C (day/night), with a photoperiod of 14 h/10 h (day/night), light intensity of 150 $\mu mol \cdot m^{-2} \cdot s^{-1}$, and relative humidity of 70–80%. Six seedlings were planted in each pot with three replicates. After 14 days of Cd stress, samples were collected and immediately cryopreserved in −80 °C liquid nitrogen for further analysis.

### 2.2. Measurement of Growth Parameters

The fresh and dry weight of cabbage leaves and roots, stem diameter, plant height, and root length were measured after 14 days of Cd treatment. The stem diameter, plant height, and root length of the cabbage were measured with calipers. The fresh weight was determined after washing the seedlings in distilled water, and the dry weight was determined by drying the plants to a constant weight at 65 °C.

### 2.3. Determination of Root Activity

Briefly, 0.5 g of root sample and 5 mL of phosphate buffer (60 mM, pH 7.0) were added to the tube and reacted at 37 °C for 2.5 h, then, 2 mL of 1 M sulfuric acid was added to the tube to stop the chemical reaction. The roots were crushed in 3–4 mL of ethyl acetate with a mortar and pestle. The red crushed liquid was then transferred to the test tube, made up to 10 mL with ethyl acetate, where the absorbance at 485 nm was recorded and the standard curve was plotted. A root activity determination was performed according to the method described by Luo et al. [19,20].

### 2.4. Determination of Chlorophyll Content

The leaf sample (0.5 g) was placed in 10 mL of 95% ethanol at room temperature, extracted in the dark for 24 h, and the supernatant centrifuged at 5000× $g$ for 10 min. The absorbance values were measured with a spectrophotometer at 649 nm and 665 nm to calculate the contents according to the method of Wintermans and De-Mots [21]. The V values indicate the dissolved volume of the solution determined, the fresh weight (FW) values indicate the fresh weight of the leaf sample, and 1000 indicates 1 L = 1000 mL.

$$Chl\ a = 13.95 \times OD665 - 6.88 \times OD649$$

$$Chl\ b = 24.96 \times OD649 - 7.32 \times OD665$$

$$Chl\ a + b\ (mg\ g^{-1}\ FW) = (Chl\ a + b \times V)/FW \times 1000$$

### 2.5. Determination and Analysis of Leaf Stomata Parameters

Three cabbage seedlings were randomly selected for each treatment, and the applying of the actinic exchange parameters of the fourth fully expanded leaf were determined from 9:00 to 10:00 using the Plant Photosynthesis Analyzer 6800 (LI-6800, LI-COR Corporation of the United States, the United States). Before determination, the seedlings were adapted in darkness for at least 30 min. The fourth functional leaf of seedlings was selected for the determination. By applying a saturation pulse under 2700 $\mu mol\ m^{-2}\ s^{-1}$, the fluorescence parameters of the minimum fluorescence (Fo′) and the maximum fluorescence yield (Fm) were obtained from the dark-adapted leaves. The actinic light was adjusted to 81 $\mu mol\ m^{-2}\ s^{-1}$, the leaves were light-adapted for 5 min, and opened every 20 s lasting for 0.8 s. By applying the actinic light, the indexes such as the minimum fluorescence (Fo′), the steady chlorophyll fluorescence (Fs), and the maximum fluorescence yield (Fm′) could

be calculated. The actual photosynthetic efficiency (Fv/Fm) was calculated as described by Genty et al. [22]. The specific computational formulas were as follows:

$$Fv/Fm = (Fm' - Fs)/Fm'$$

### 2.6. Determination and Analysis of Leaf Chlorophyll Fluorescence Parameters

We randomly selected three cabbage seedlings from each treatment and measured the relevant parameters of the fourth fully unfolded leaf using a portable, modulated chlorophyll fluorescence analyzer (PAM2500, WALZ, Germany), including the maximum photochemical efficiency (Fv/Fm), the actual photochemical efficiency Y(II), the non-photochemical quenching (NPQ), and the quantum yield for the regulation of energy dissipation in the photochemical quenching (qP).

### 2.7. Determination of Se and Cd Contents

The root and leaf tissues were dried at 65 °C until the sample reached a constant weight, then, the dried plant samples (0.5 g) were ground to powder, mixed with nitric acid/perchloric acid (4:1, $v/v$), and boiled at 220 °C until clear and used for constant volume determination. The Cd content was determined by a flame atomic absorption spectrophotometer (ZA3000, Hitachi, Japan) [23] and the Se content by a liquid phase atomic fluorescence photometer (LC-AFS-8530, Beijing Haiguang Instrument Co., Ltd., China) [24].

### 2.8. Chloroplast Ultrastructure Analysis

The fourth fully expanded leaf (approximately 1 mm$^2$) of cabbage from each treatment was fixed in 2.5% ($v/v$) glutaraldehyde (0.1 M phosphate buffer, pH 7.2) overnight, soaked in 1% ($v/v$) osmic acid, and washed with 0.1 M PBS (pH 7.4) after 1–2 h. Then, the sample was dehydrated in 30–50–70–80–95–100% alcohol sequentially, dehydrated in 100% ethanol for 15 min 3 times, and immersed in Epon-812 epoxy resin at 60 °C overnight. Finally, the samples were semi-sectioned, stained, rinsed with dye solution, fixed in copper mesh, scanned by transmission electron microscopy, and photographed for observation [25].

### 2.9. Determination and Analysis of Electrolyte Leakage

The fourth fully unfolded leaf of the seedling under each treatment was selected to determine the electrical conductivity and the specific measurement method was taken from Bajji [26]. 0.5 g of cabbage leaves were measured in a test tube containing 10 mL of deionized water (E Ca). The contents were heated in a water bath at 50–60 °C for 25 min and the EC values were recorded (E Cb). The contents were then boiled at 100 °C for 10 min and the EC was again recorded (E Cc). The electrolytic leakage was calculated using the formula:

$$Electrolyte\ leakage\ (\%) = (E\ Cb - E\ Ca)/E\ Cc \times 100$$

### 2.10. Determination and Analysis of MDA and Proline Content

Half a gram (0.5 g) of cabbage leaves were weighed, 5 mL of 0.1% trichloroacetic acid (TCA) was added, crushed completely in a pre-cooled mortar, and centrifuged at $4000 \times g$ for 15 min at 4 °C. The supernatant was the liquid to be determined. 1 mL of the supernatant was absorbed and 4 mL of a 20% TCA mixture containing 0.65% ($w/v$) TBA were added. The mixture was reacted in boiling water for 15 min, immediately cooled, and centrifuged again to determine the light absorption at 450, 532 and 660 nm. V is the total volume of extract (mL), V1 is the volume of sample extract added to the solution to be tested (mL), and V2 is the total volume of solution to be tested, 4 mL.

$$MDA\ (\mu mol/g) = [6.45(A532 - A600) - 0.56 \times A450] \times V2 \times V/m \times V1 \times 1000 \quad (1)$$

Half a gram (0.5 g) of cabbage leaves were weighed and pulverized with liquid nitrogen, 7.5 mL of 3% sulfosalicylic acid was added to regrind completely, and then centrifuged at 4 °C for 20 min (11,000× $g$). 2 mL of the supernatant was removed and

mixed by adding 2 mL of ninhydrin and 2 mL of 3.5% acetic acid, and the mixed solution was boiled at 100 °C for 1 h. After cooling, 4 mL of toluene was added to the reaction mixture, and the absorbance value was measured at 520 nm [27].

### 2.11. $O^{2-}$ and $H_2O_2$ Histochemical Staining and Content

Half a gram (0.5 g) of fresh leaf tissue was ground with 5 mL of 50 mM phosphate buffer (pH 7.8), and the mixture centrifuged at $10,000 \times g$ for 20 min. The $O^{2-}$ content was measured in the supernatant and calculated from the standard curve based on the sodium nitrite and expressed as $\mu mol \cdot g^{-1} \cdot min^{-1}$ FW. To determine the $H_2O_2$ content, 0.5 g of fresh leaf tissue were ground in an ice bath with 5 mL 100% cold water acetone, centrifuged at $10,000 \times g$ 4 °C for 20 min, and the supernatant was collected immediately for the $H_2O_2$ analysis. The concentration of $H_2O_2$ was measured according to the standard curve and expressed as $\mu mol \cdot g^{-1}$ FW.

The nitrogen blue tetrazole (NBT) method was used for the $O^{2-}$ staining analysis of the cabbage leaves: The leaves were soaked in 0.5 $mg \cdot mL^{-1}$ NBT solution (pH 7.8) in the dark for 1 h, then removed, rinsed 3 times with distilled water and then heated with 95% ethanol for 10 min for decolorizations. The blue spots on the leaves indicated the production of $O^{2-}$. The 3,3-diaminobenzidine (DAB) staining method was used to stain the tissue of cabbage leaves for $H_2O_2$: The leaves were soaked in 1 $mg \cdot mL^{-1}$ DAB solution (pH 7.8) at 25 °C for 5 h, then decolored with 95% ($v/v$) ethanol every 20 min at 80 °C until the green background was completely removed from the leaves. Red-brown spots on the leaves indicate $H_2O_2$ production.

### 2.12. Antioxidant Enzyme System Determination and Analysis

The 0.5 g leaf sample was weighed and ground to a homogenate with 3 mL of phosphate buffer (pH 7.0) consisting of 0.2 mM EDTA and 2% ($w/v$) PVP. The homogenate was centrifuged at 4 °C for 20 min at $13,000 \times g$, and the supernatant was the enzyme liquid to be measured, which was used for the determination of the SOD and POD enzyme activities. APX assay: 3 mL of the reaction mixture contained 50 mM phosphate buffer saline (PBS) (pH 7.0), 9 mM ASA, 12.5 mM $H_2O_2$, and 100 μL of enzyme solution, and the enzyme activity was calculated based on the change in absorbance at 290 nm every 1 min. The SOD activity was estimated by measuring its ability to inhibit the photochemical reduction of NBT Giannoplitis and Ries [28]. POD activity assay was determined according to the method of Mao et al. [29].

The level of GSH was determined using the Solarbio BC1175 kit for the content of the reduced glutathione.

The determination was carried out by means of the Solarbio BCO220 Ascorbate Peroxidase Kit.

### 2.13. Statistical Analysis

All data were analyzed using a one-way ANOVA followed by Duncan multiple range tests. A $p$ value < 0.05 was considered significant. Values are presented as mean ± standard deviation (SD) of at least three independent experiments with three replicates for each treatment ($p \leq 0.05$).

## 3. Results

### 3.1. Exogenous Se Can Promote the Growth of Cabbage Seedlings under Cd Stress

Compared with the control, Cd stress caused significant leaf chlorosis and curling, and inhibited plant and root growth. Exogenous spraying of Se improved the growth of the cabbage seedlings and alleviated the symptoms of leaf yellowing (Figure 1). Under normal conditions, the exogenous Se increased the growth of the cabbage seedlings. Compared with the control, the fresh weight (FW) and dry weight (DW) of the leaves and roots, plant height, stem diameter, and the root length of the cabbage seedlings were significantly decreased under Cd stress. Compared with Cd treatment alone, the application of exogenous Se under

Cd stress could significantly increase the FW and DW of leaves and roots, plant height, stem diameter, and the root length of the cabbage seedlings by 30.14%, 34.36%, 21.21%, 60%, 18.30%, 16.67% and 28.20%, respectively (Figure 2A–G). In addition, compared with the control group, root activity was significantly decreased under Cd stress, while compared with Cd stress alone, root activity of the cabbage seedlings was significantly increased by 82.92% when the exogenous Se was applied under Cd stress. The results showed that the exogenous Se could promote the growth of the cabbage seedlings under Cd stress by increasing root activity (Figure 2I). Compared with the control group, the total chlorophyll content of the cabbage seedlings decreased significantly under Cd stress alone but increased when treated with the exogenous Se alone. Compared with the treatment of Cd stress alone, the total chlorophyll content of the cabbage seedlings increased by 55.56% when the exogenous Se was applied under Cd stress, indicating that the exogenous Se could alleviate the decrease of photosynthetic pigment content in the cabbage seedlings under Cd stress (Figure 2H).

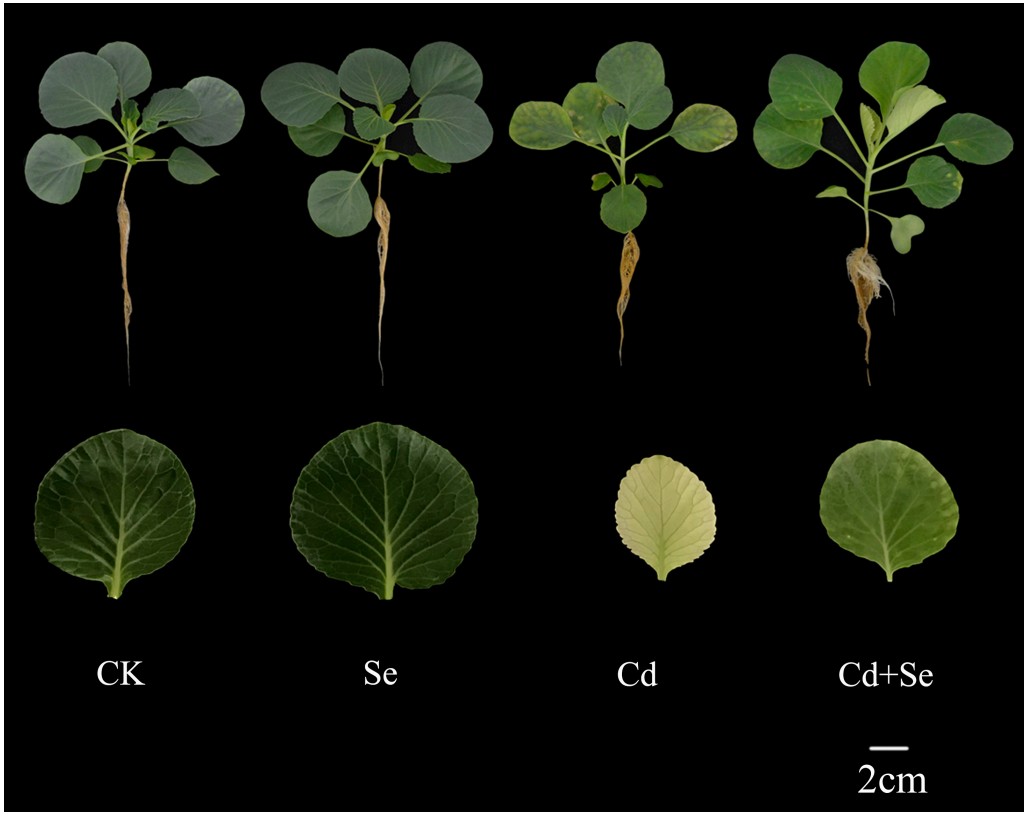

**Figure 1.** Effect of exogenous Se on plant and leaf morphology of cabbage seedlings under Cd stress.

### 3.2. Effects of Exogenous Se on Stomatal Parameters of Cabbage Seedlings under Cd Stress

Compared to the control, Cd treatment significantly reduced the stomatal exchange capacity of the cabbage leaves. When compared with Cd treatment alone, the exogenous Se increased the content of the net photosynthetic rate (Pn) 65.4%, stomatal conductance (Gs) 32.5%, and transpiration rate (Tr) 72.84%, respectively, in the cabbage leaves under Cd stress, while decreasing the content of intercellular $CO_2$ concentration (Ci) 14.59%. When compared with the control, no significant difference was found in Pn, Gs, Tr, and Ci (Figure 3A–D), but the effect was alleviated by the application of the exogenous Se. When treated with Se alone, the contents of Pn, Gs, and Tr in the leaves of the cabbage seedlings were not significantly different from the control group, while the Ci was significantly increased.

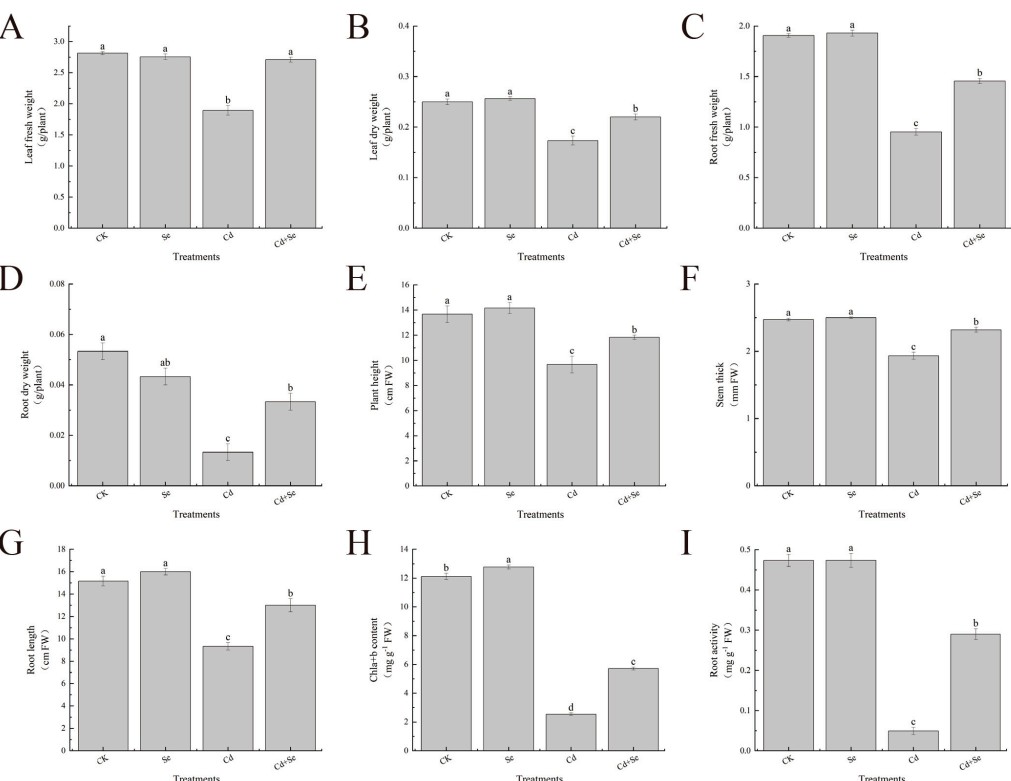

**Figure 2.** Effect of exogenous Se on the growth of cabbage seedlings under Cd stress. (**A**) leaf fresh weight, (**B**) leaf dry weight, (**C**) root fresh weight, (**D**) root dry weight, (**E**) plant height, (**F**) stem thick, (**G**) root length, (**H**) Chla + b content, and (**I**) root activity. Values are means of three replicates ± SD. Different letters indicate significant difference at $p \leq 0.05$.

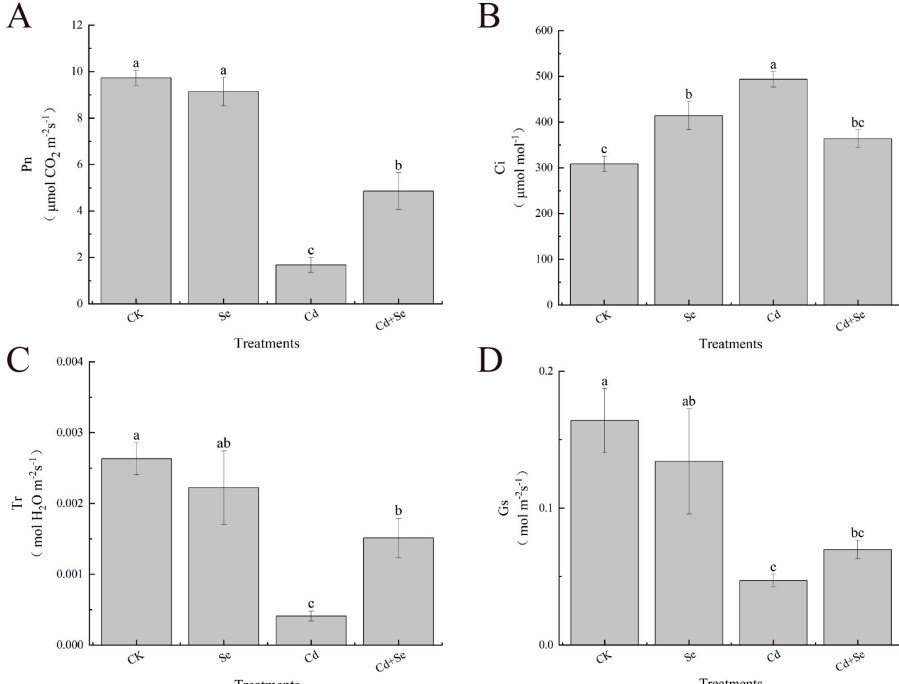

**Figure 3.** Effects of exogenous Se on stomatal parameters of cabbage seedlings under Cd stress. (**A**) net photosynthetic rate (Pn), (**B**) intercellular $CO_2$ concentration (Ci), (**C**) transpiration rate (Tr), (**D**) stomatal conductance (Gs). Values are means of three replicates ± SD. Different letters indicate significant difference at $p \leq 0.05$.

*3.3. Effects of Exogenous Se on Chlorophyll Fluorescence Parameters of Cabbage Seedlings under Cd Stress*

Cd stress alone inhibited chlorophyll synthesis decreased the stomatal exchange parameters and reduced the PSII activity in the leaves of the cabbage seedlings. Compared with the control group, the maximum quantum yield of PSII (Fv/Fm), actual photochemical efficiency of PSII Y(II), and photochemical quenching coefficient (qP) of the leaves of the cabbage seedlings were reduced by 78.5%, 68.51%, and 59.55%, respectively. However, the exogenous Se applied under Cd stress can significantly reduce the Cd-induced decreases in Fv/Fm, Y(II), and qP. The above results indicated that the application of the exogenous Se to cabbage seedlings could significantly alleviate Cd stress and increase the maximum photochemical efficiency of the leaves under Cd stress. On the contrary, the non-photochemical quenching (NPQ) value of the leaves of the cabbage seedlings treated with Cd alone was the highest, but this phenomenon was alleviated by spraying the exogenous Se. In addition, there was no significant difference between the leaves of the cabbage seedlings and the control treatment when the exogenous Se was sprayed alone (Figure 4A–D).

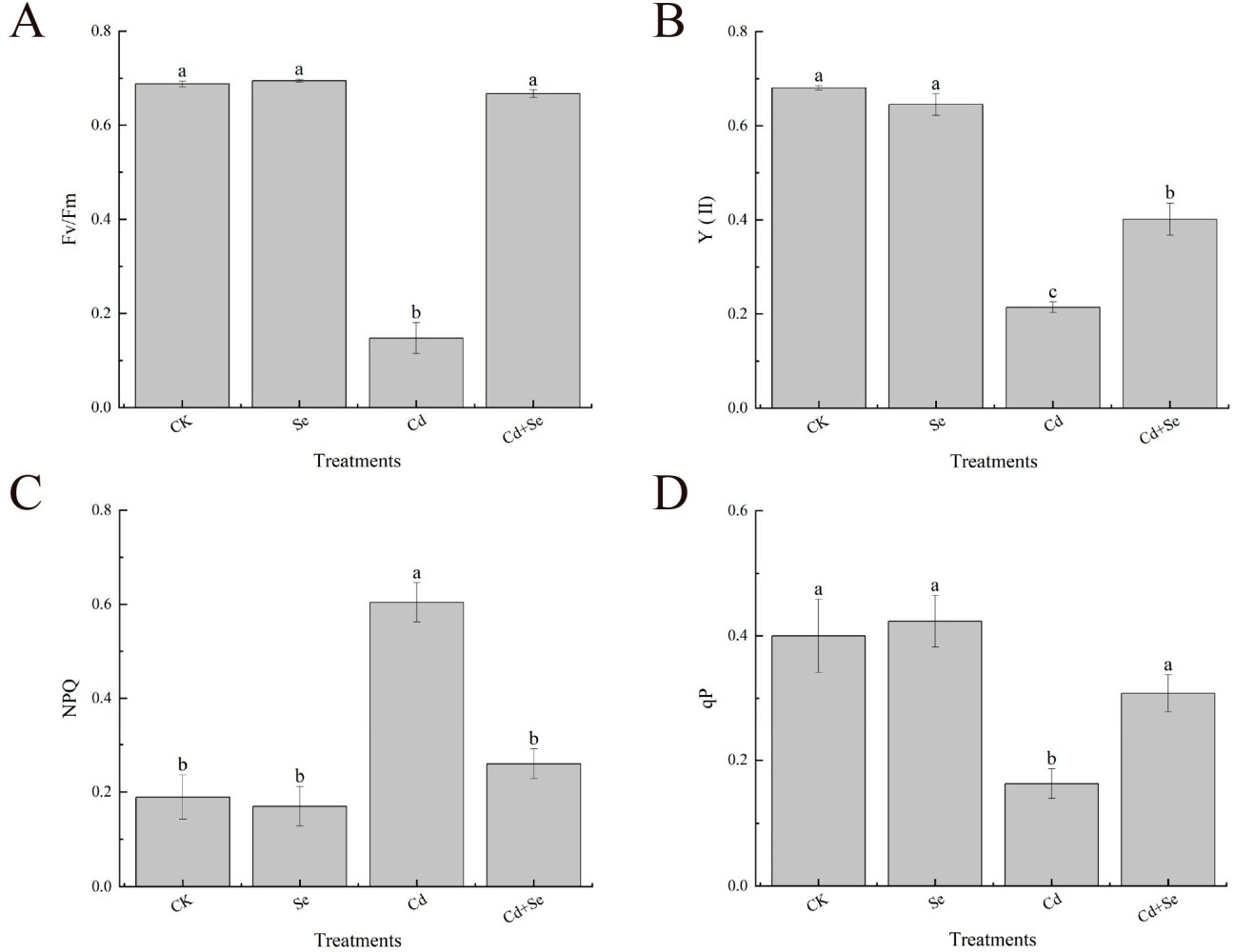

**Figure 4.** Effects of exogenous Se on chlorophyll fluorescence parameters of cabbage seedlings under Cd stress. (**A**) maximum quantum yield of PSII (Fv/Fm), (**B**) actual photochemical efficiency of PSII [Y(II)], (**C**) non-photochemical quenching (NPQ), (**D**) photochemical quenching coefficient (qP). Values are means of three replicates ± SD. Different letters indicate significant difference at $p \leq 0.05$.

*3.4. Exogenous Se Can Enhance the Tolerance of Cabbage Seedlings under Cd Stress*

The results showed that the Cd content in the leaves and roots was highest under Cd treatment alone. The exogenous application of Se significantly reduced the Cd content in

the leaves and roots of the cabbage seedlings, and compared with Cd treatment alone, the Cd content in the leaves and root tissues of the cabbage seedlings was significantly reduced to 73% and 76%, respectively (Figure 5A,B). The Cd content in the leaves was significantly lower than that in the roots after the exogenous Se treatment.

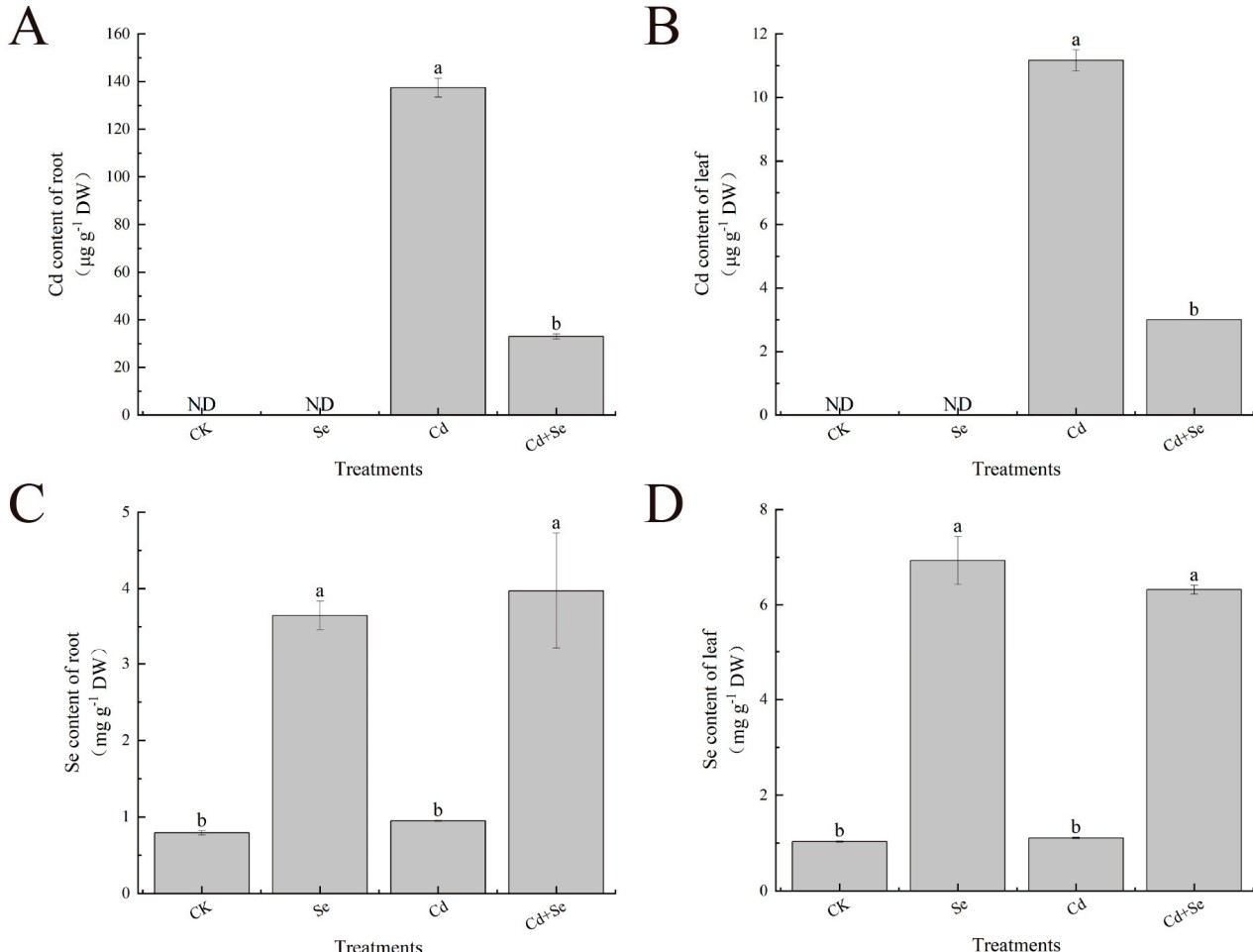

**Figure 5.** Effects of exogenous Se on tolerance of cabbage seedlings under Cd stress. (**A**) Cd content of root, (**B**) Cd content of leaf, (**C**) Se content of root, (**D**) Se content of leaf. Values are means of three replicates ± SD. Different letters indicate significant difference at $p \leq 0.05$.

*3.5. Observation of Chloroplast Ultrastructure in Leaves of Cabbage Seedlings*

The internal structure of chloroplasts in the leaves of the cabbage seedlings was further observed to determine whether the exogenous application of Se can reduce chloroplast damage caused by Cd stress. The results showed that the control group had good morphology, complete cell structure, uniform thickness of the cell wall (CW), tight adhesion between the cell membrane (CM) and cell wall, and no obvious plasma wall separation. Chloroplasts (Ch) were abundant, mostly in short fusiform form, a small amount of starch granules (SG) and granular lamellae (GL) were found in the interior, and a small amount of osmophilic granules (OG) were found in the leaf green (Figure 6A–C). The seedlings treated with Se alone were similar to the control group in that the cell structure was intact, the cell wall (CW) was intact, the membrane (CM) was closely attached to the cell wall and no obvious plasma wall separation was observed. The number of chloroplasts (Ch) was normal, and most were short fused with a small amount of starch granules (SG), a regular arrangement of grana lamellae (GL), and a fuzzy structure. The mitochondria (M) are elliptical, and the vacuole (V) membrane is intact (Figure 6D–F). Cd alone caused partial disintegration of the cell structure, the cell wall (CW) remained intact, the cell membrane (CM) was

damaged in a small area, the cell matrix dissolved, and the plasmalemma wall separated. The number of chloroplasts (Ch) decreased significantly, some morphological structures were damaged or disappeared, a small number of chloroplasts were spherical, the grana lamella (GL) was reduced, the structure was unclear, a small number of chloroplasts were expanded, and a small number of osmophilic particles (OG) without uniform size were found in the chloroplasts. In addition, the small number of osmophilic particles in the leaf green were different in size compared to the control group, and no obvious starch particles were found (Figure 6G–I), indicating that the photosynthetic organs in the leaves of the cabbage seedlings were damaged under Cd stress. After the exogenous application of Se, chloroplast morphology was obviously restored, the cell structure was mostly present, the cell wall (CW) was intact, the cell membrane (CM) was damaged and disappeared, and the plasmalemma wall was separated. The number of chloroplasts (Ch) was normal, free, and most of them were in good condition. The arrangement of grana lamellae (GL) was regular and some of them were moderately dilated. There were more starch granules (SG) in the chloroplasts with larger volume. A small number of free osmophilic particles (OG) could be seen in the cells (Figure 6J–L), indicating that the exogenous Se application alleviated the damage caused by the Cd stress on the chloroplasts to an extent, improved the stability of the inner capsule membrane, and alleviated the damage caused by the Cd stress on the photosynthetic organs.

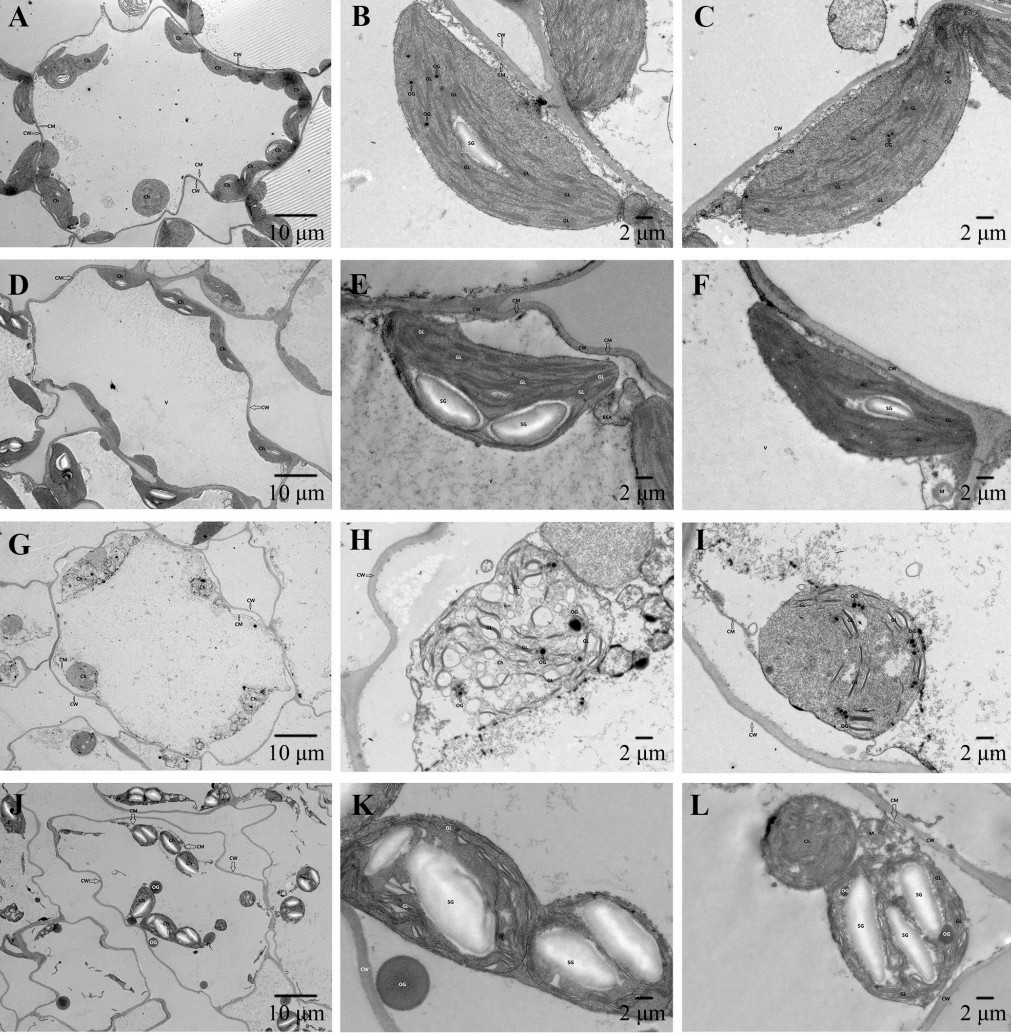

**Figure 6.** Observation on chloroplast ultrastructure of cabbage seedling leaves. CK. (**A–C**), Se. (**D–F**), Cd. (**G–I**), Cd + Se. (**J–L**). Abbreviation: cell wall (CW), cell membrane (CM), chloroplast (Ch), osmiophilic globule (OG), grana lamellae (GL), mitochondria (M), vacuole (V), starch granules (SG).

### 3.6. Effects of Exogenous Se on Reactive Oxygen Species Accumulation and Leaf Membrane Esterification of Cabbage Seedlings under Cd Stress

When the Cd was treated alone, the leaves of the cabbage seedlings were stained by the diaminobenzidine (DAB) method and a nitroblue tetrazolium (NBT) chloride blue reagent, respectively. The leaves of the cabbage seedlings were stained blue and yellow, respectively, and the degree of staining was worse than that of the control, indicating that the Cd treatment alone could damage the integrity of the cell membrane of the cabbage seedlings and aggravate membrane lipid peroxidation of the leaves. There was no significant difference between the Se treatment alone and the control. The degree of blue and yellow staining of the leaves of the cabbage seedlings was reduced after the exogenous application of Se under Cd stress, indicating that the exogenous application of Se can alleviate the oxidative damage of leaves of cabbage seedlings under Cd stress (Figure 7A,B).

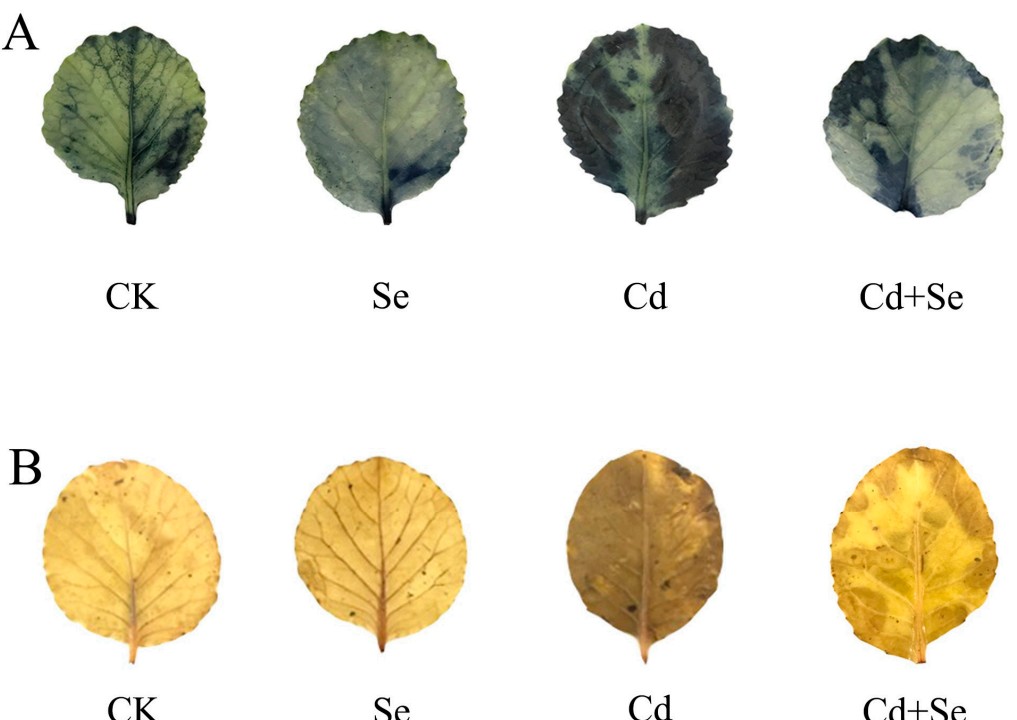

**Figure 7.** Effects of exogenous Se on diaminobenzidine (DAB) and nitroblue tetrazolium (NBT) staining of leaves of cabbage seedlings under Cd stress (**A**,**B**).

The degree of oxidative damage in the leaves under Cd stress is related to the excessive accumulation of ROS. The levels of $O^{2-}$ and $H_2O_2$ in the leaves of the cabbage seedlings were measured and it was found that the levels of $O^{2-}$ and $H_2O_2$ in the leaves of the Cd-stressed cabbage seedlings accumulated most when treated with Cd alone. There was no significant difference between the Se treatment alone and the control. However, compared with Cd stress alone, the exogenous application of Se treatment under Cd stress reduced the content of $O^{2-}$ and $H_2O_2$ in the leaves of the cabbage seedlings to 13.63% and 40.83%, respectively (Figure 8A,B). In addition, Cd alone had the greatest effect on the electrical conductivity, proline, and malondialdehyde contents of the cabbage leaves. Compared with the Cd treatment alone, the exogenous application of Se under Cd stress significantly reduced the electrical conductivity and malondialdehyde content of the cabbage seedlings to 50.63% and 33.87%, respectively, and further increased the proline content of the cabbage seedlings by 80.3%. There was no significant difference between the Se addition alone and the control group (Figure 8C–E).

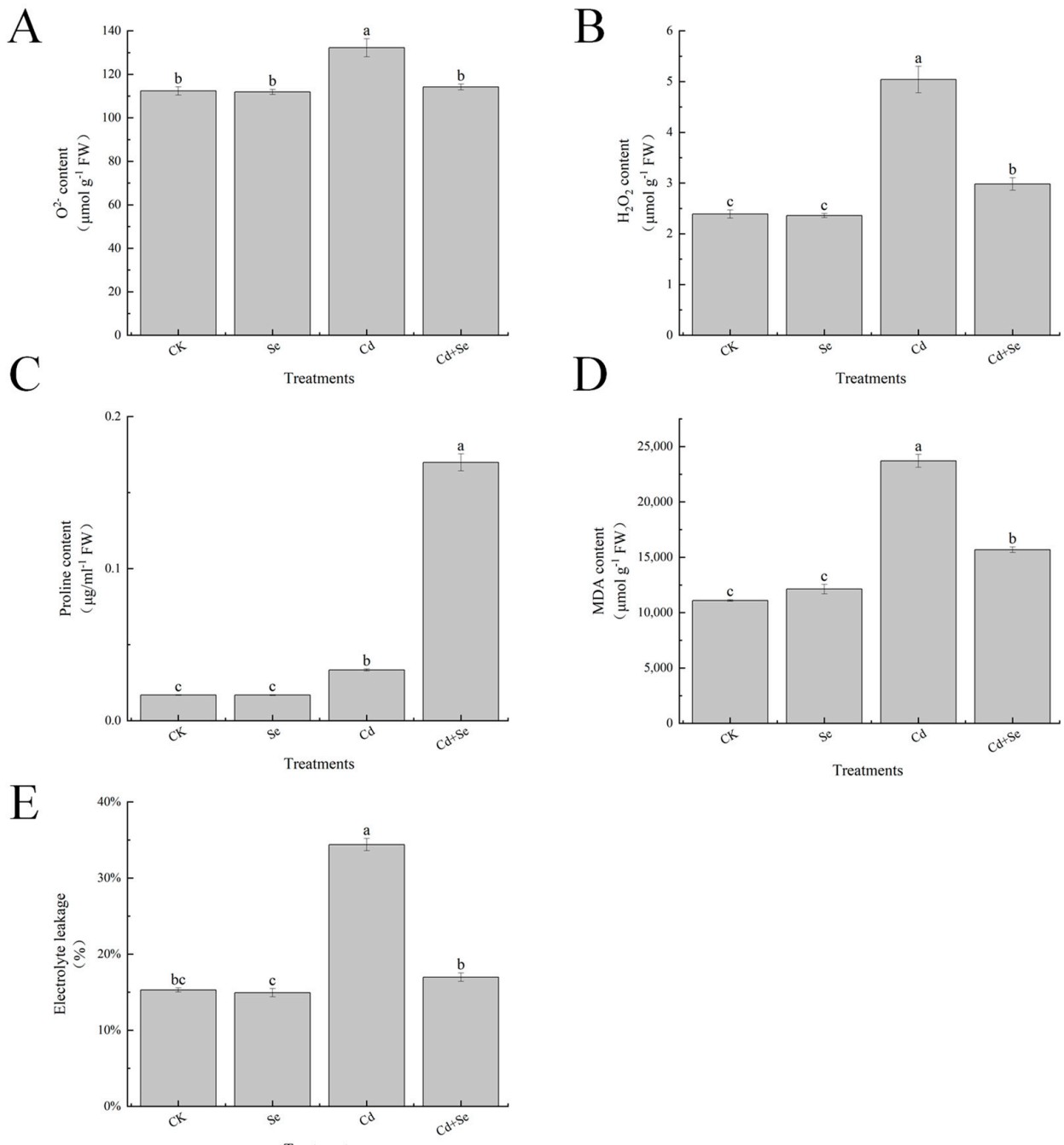

**Figure 8.** Effects of exogenous Se on $O^{2-}$ content, $H_2O_2$ content, Proline content, MDA content, Electrolyte leakage of cabbage seedlings under Cd stress (**A–E**). Values are means of three replicates ± SD. Different letters indicate significant difference at $p \leq 0.05$.

### 3.7. Effects of Exogenous Se on Reactive Oxygen Species Accumulation and Leaf Membrane Esterification of Cabbage Seedlings under Cd Stress

The activities of superoxide dismutase (SOD), peroxidase (POD), ascorbate peroxidase (APX), and the content of reduced glutathione (GSH) in the leaves of the cabbage seedlings were determined to elucidate the role of the exogenous Se in the antioxidant system under Cd stress. The activities of SOD, POD, GSH and APX in the leaves of the cabbage seedlings treated with the Se alone were similar to the control and the activity of APX was increased. The activities of SOD, POD, APX, and GSH were significantly higher than the control when treated with Cd alone. However, when treated with the combination of Cd and Se, the

activities of SOD, POD, APX, and GSH were further increased by 23.36%, 52.28%, 36.54%, and 8.33%, respectively (Figure 9A–D). The exogenous Se could alleviate the Cd-induced damage, indicating that Se could alleviate the Cd-induced oxidative stress by regulating the antioxidant activity of the leaves.

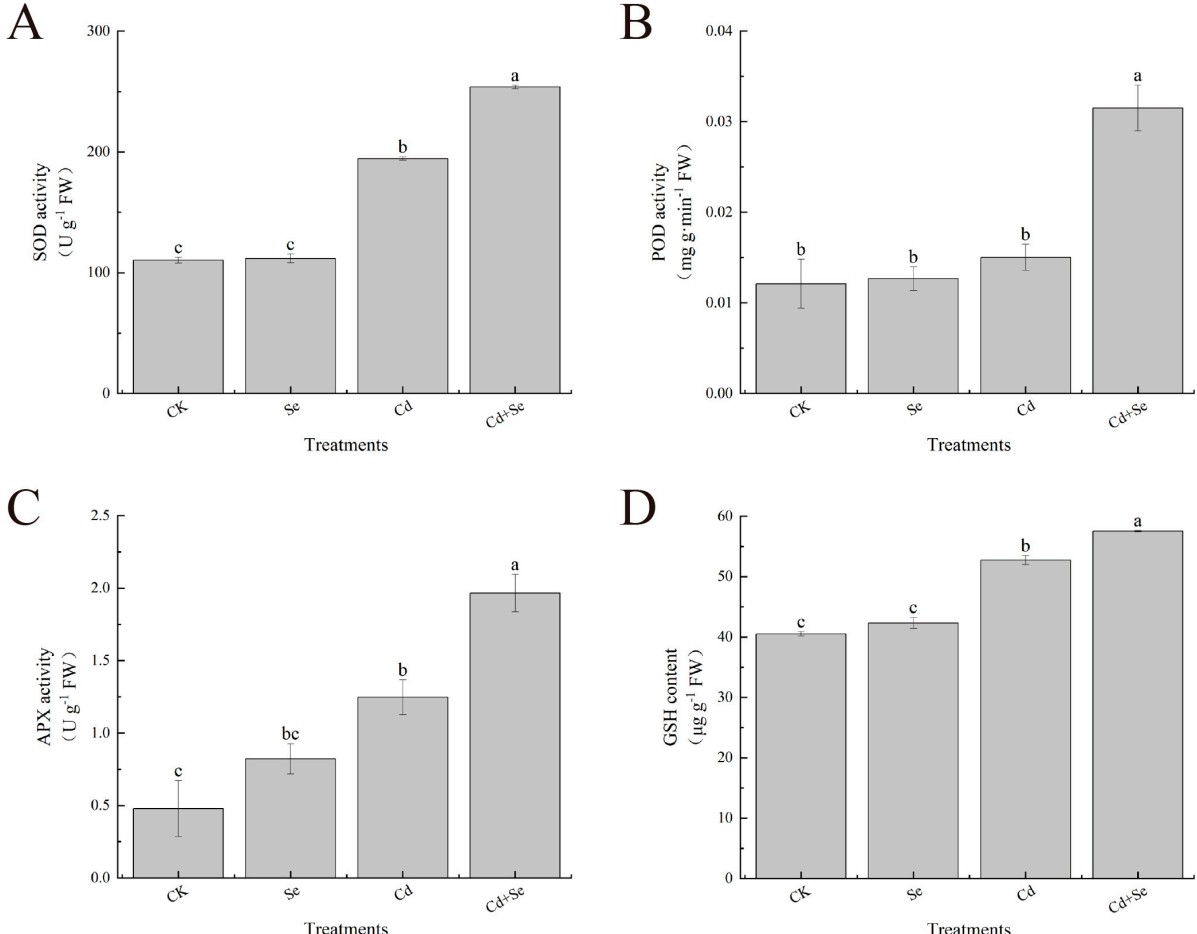

**Figure 9.** Effects of exogenous Se on antioxidant enzyme activities of cabbage seedlings under Cd stress. (**A**) SOD activity, (**B**) POD activity, (**C**) APX activity, (**D**) GSH content. Values are means of three replicates $\pm$ SD. Different letters indicate significant difference at $p \leq 0.05$.

## 4. Discussion

Heavy metal contamination of the soil is a global problem and poses a serious threat to public health [30–32]. Cd is highly toxic to plants [33], and the accumulation of Cd in edible parts of crops is one of the main threats to human health. Reducing the damage of abiotic stress to plants by applying exogenous substances has gradually become the focus of research [34]. As one of the essential elements for animals and humans, Se has been proven to reduce the toxicity of Cd, lead Pb, and arsenic s to plants [35–37]. Therefore, the application of exogenous Se to alleviate Cd stress in cabbage seedlings is of great significance for agricultural production.

In this study, we analyzed the physiological and biochemical parameters of cabbage seedlings regulated by the exogenous Se application under Cd stress. Under Cd stress, the cabbage plants showed obvious symptoms such as leaf rolling, yellowing, inhibition of root elongation, and a reduction of biomass (Figure 1), which was consistent with studies in *Brassica napus*, mustard, tomato, and cucumber [18,38–40]. On the contrary, the exogenous application of Se could alleviate these phenomena, and the exogenous application of Se promoted the growth of the cabbage seedlings under Cd stress, which was reflected in the increase of plant height, stem diameter, and biomass (Figure 2A–G). This was consistent

with the research results of Filek M and Qi W. [41,42], which indicated that Se could reduce the growth inhibition of the cabbage seedlings under Cd stress (Figure 2I). The root activity may reflect the growth status of plants, and the activities of the roots treated with Cd alone were significantly reduced, while they were apparently restored after the exogenous application of Se, which was consistent with the results of Jiang J's study [43]. These results indicated that Se could enhance root vitality by improving the efficiency of root water and nutrient uptake to promote plant growth under Cd stress.

Our research showed that the Cd content in the leaves and roots of the cabbage seedlings was significantly increased under Cd stress and was higher in the roots than in the leaves (Figure 5A), indicating that Cd is trapped in the roots and a small amount is subsequently transported to the leaves. This suggest that the exogenous Se is antagonistic to Cd. The Cd content in the leaves was significantly lower than that in the roots after the exogenous Se treatment, indicating that the exogenous Se could inhibit the transfer of Cd from the underground to the aboveground part. This is done by regulating the Cd content in different parts of the cabbage seedlings, thus improving the tolerance of the cabbage seedlings to Cd stress. The results are consistent with those of previous studies that state that Cd accumulates more in the roots than in the leaves. However, the Cd content in the leaves and roots was reduced after the exogenous Se application, which is similar to the results of the study by Li L et al. in mustard. In conclusion, an exogenous Se application has a positive effect on reducing the Cd content in cabbage seedlings, and the difference in the positive effect of the Se on different plant parts may be related to the accumulation of Cd in the upper part of the plant.

Cd stress can lead to a decrease in photosynthetic pigment content in the plants, which has been reported in cabbage, broccoli, pepper, and other plants [43–45]. Our results showed that the Cd treatment alone resulted in a significant decrease in the total chlorophyll content, and at the same time, the Cd stress also significantly decreased the gas exchange characteristics such as Pn, Gs, and Tr, and increased Ci (Figure 3A–D). We speculated that Cd stress led to stomatal closure, reduced mesophyll conductance and $CO_2$ fixation, and ultimately photoinhibition. Se can promote growth and development of the plants under various abiotic stresses. In this study, the exogenous Se application effectively alleviated the the Cd-induced decrease in chlorophyll content and photosynthetic characteristics, which was consistent with previous studies [46].

Chlorophyll fluorescence can reflect the photosynthetic efficiency and photosynthetic inhibition of plants under heavy metal stress. In this study [47], the Cd treatment significantly reduced the Fv/Fm, Y(II), and qP parameters, but increased the NPQ value (Figure 4A–D), indicating that the PSII under Cd stress had an imbalance between light energy absorption and photosynthetic electron transfer [48,49]. However, the exogenous Se can significantly increase the chlorophyll content and the photosynthetic capacity and can reduce the Cd stress-induced photoinhibition by increasing Fv/Fm, Y(II), and qP [50,51].

Chloroplasts, as the sites of photosynthesis in plants, are involved in the regulation of various physiological responses and are also organelles sensitive to Cd stress. In this study, the ultrastructure of the leaf mesophyll cells of the cabbage seedlings was significantly damaged under the Cd stress, as shown by the chloroplast deformation and damage, poor lamellar arrangement, reduction of starch granules, and the appearance of osmophilic particles, which was consistent with previous studies [52,53]. The results showed that the Cd stress significantly decreased the photosynthetic rate, chlorophyll content, and starch granule accumulation in the cabbage chloroplasts, which was consistent with the results of H. Y. Sun et al. Therefore, we speculate that the structural changes in the chloroplasts under Cd stress may be the direct cause of the damage to the photosynthetic system and chlorophyll degradation, which ultimately leads to the inhibition of photosynthesis.

However, the exogenous application of Se restored the damage to the chloroplast structure, increased the number of starch granules in the cabbage leaves, and effectively improved the stability and integrity of the chloroplast structure (Figure 6). This indicates that the Se can protect the growth of the cabbage seedlings and alleviate the damage to the

chloroplast structure under Cd stress, thus promoting photosynthesis, which played a key role in alleviating the Cd stress in the cabbage seedlings.

To investigate the effect of the exogenous Se application on the scavenging of the ROS in the seedling leaves, we examined the levels of $O^{2-}$ and $H_2O_2$ in the leaves of the cabbage seedlings. The results showed that Cd caused a significant increase in the production of $O^{2-}$ and $H_2O_2$, resulting in excessive ROS production (Figures 7 and 8), which was consistent with the results of He S et al. [54]. However, the exogenous application of Se minimized the production of ROS induced by Cd stress, possibly due to its function of protecting cells from the Cd-induced oxidative damage.

At the same time, the levels of MDA, proline, and electrolyte leakage are used as important indicators of lipid peroxidation in plants to reflect the degree of membrane damage [55]. Our results showed that the Cd treatment significantly increased the generation of MDA, proline, and electrolyte leakage in the leaf tissues compared to the control (Figure 8D). According to the histochemical detection, the staining degree of leaves of the cabbage seedlings treated with Cd alone was higher, indicating that the Cd stress caused a severe cellular REDOX imbalance and oxidative damage, increased lipid peroxidation, and excessive ROS accumulation. It can be concluded that the oxidative damage caused by Cd in plants is the reason for the destruction of the cell membrane, inhibition of nutrient absorption, and enhanced lipid peroxidation [56].

Plants induce resistance to biotic and abiotic stresses by producing antioxidant enzymes, and oxidative damage is closely related to their antioxidant defense mechanisms. The results showed that the activities of the SOD, POD, APX, and GSH of the cabbage seedlings were increased under Cd stress, while the exogenous application of Se further improved the activity of the cabbage seedling leaves (Figure 9), indicating that Se could significantly reduce the oxidative damage caused by Cd stress by enhancing the antioxidant response. Our results were consistent with those of Qi W et al. who confirmed that Cd-stressed *Brassica napus* plants could enhance the activity of antioxidant enzymes, that the enzyme activity was further improved after the Se application, and that the antioxidant enzymes were closely related to Cd tolerance [40]. These results suggest that the exogenous Se may induce REDOX homeostasis by activating both enzyme-supported and non-enzyme-supported antioxidant systems, thereby protecting the cabbage seedlings from Cd stress.

## 5. Conclusions

In this study, we found that exogenous Se can effectively alleviate the stress of Cd on the growth, and the physiological and biochemical characteristics of cabbage seedlings, inhibit the transport of Cd from the root to the leaf, and restore the negative effect of Cd stress. The exogenous Se application is an important way to alleviate Cd stress.

Cd stress severely inhibited the growth and photosynthesis of the cabbage seedlings, increased the levels of $O^{2-}$, $H_2O_2$, MDA, electrolyte leakage, and proline, decreased the antioxidant capacity of SOD, POD, APX, and GSH, and disturbed the nutrient balance. However, the exogenous application of Se increased the biomass and photosynthetic efficiency of the cabbage seedlings and decreased the concentration of Cd, which was conducive to maintaining the growth of the cabbage seedlings and eliminating the excess ROS by upregulating antioxidant enzymes to protect the ultrastructure of chloroplasts. Therefore, the results confirmed the positive effect of the exogenous Se application on Cd stress in the cabbage seedlings, which laid the foundation for the study of an exogenous Se application to reduce Cd stress in cabbage (Figure 10).

In conclusion, this study elucidates the role and regulatory mechanism of Se under Cd stress in cabbage, which is helpful for the application of Se under Cd stress in agricultural production to promote plant growth and increase yield.

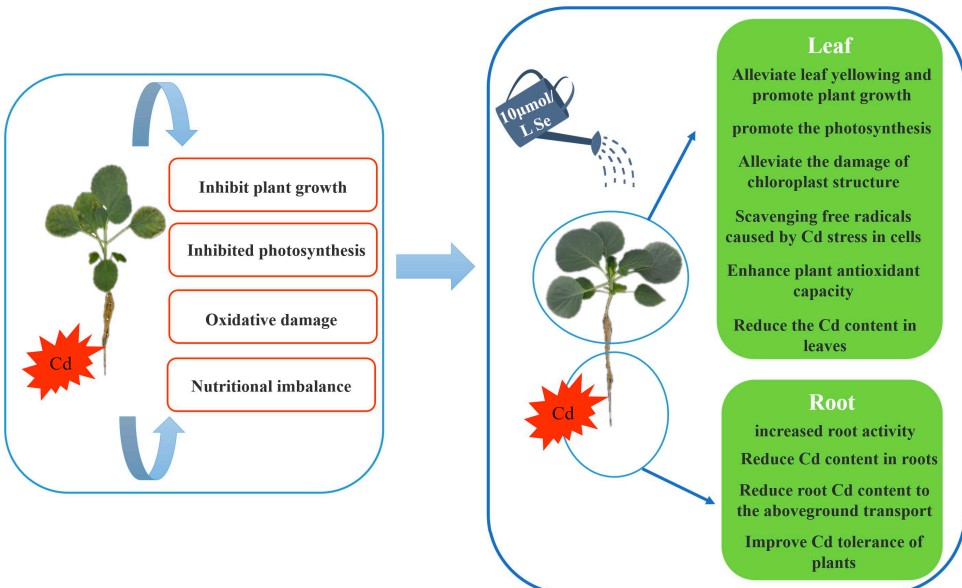

**Figure 10.** Exogenous Se has an alleviating effect on the physiological mechanism of cabbage seedlings.

**Author Contributions:** K.J. was the experimental designer and performer of the experimental study, completed the data analysis, and the first draft of the paper; Z.Z., B.W., W.W. (Wuhong Wang) and W.W. (Wenjing Wei). participated in the experimental design and analysis of the experimental results; Z.X., W.H. and D.L. were the conceptualizers and leaders of the project, and led the experimental design, data analysis, paper writing, and revision. All authors have read and agreed to the published version of the manuscript.

**Funding:** This research was funded by the National Key Research and Development Program of China (2016YFD0101702), the Special Funds Project for the Construction of National Modern Agricultural Industrial Technology System (CARS-23-G22), and the Science and Technology Program of Xi'an Municipality (20NYTX0001).

**Data Availability Statement:** Data are contained within the article.

**Acknowledgments:** We thank Ruihong Chen and Lijuan Lu (Horticulture Science Research Center, Northwest A&F University, Yangling, China) for their assistance.

**Conflicts of Interest:** The authors declare no conflict of interest.

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
