# Peer review of "Exogenous Selenium Enhances Cadmium Stress Tolerance by Improving Physiological Characteristics of Cabbage (Brassica oleracea L. var. capitata) Seedlings"

_horticulturae, doi:10.3390/horticulturae9091016_

Round 1

Reviewer 1 Report

Dear authors,

Congratulations on your manuscript, which I read with pleasure. The manuscript shows us an interesting set of results concerning the exogenous application of selenium to enhance cadmium stress tolerance of cabbage seedlings. Writing is in general efficient but there are problems in material and methods and results sections that need to be presented in a clearer way. Some typographical errors were also found (please see details below and carefully read the MS to find more). The discussion is very clear and supported by the results. However, there are several problems concerning the references/bibliography list, both in the way how it is presented in the text (please homogenize it), as well as concerning the numbers mentioned in the text which does not correspond to the numbers given in the reference list. The MS has great potential to be published but first these points should be corrected, and the questions below should be properly addressed before I can endorse it to publication.

Also, the MS should have line number to simplify the revision process. Please take this into consideration before submitting the improved version

Abstract

-       The first time that an abbreviation/acronym is given (ROS, O2-, H2O2, MDA, SOD, POD. APX and GSH) the full name should also be provided.

Keywords

-       should be provided in alphabetical order

Introduction

-       Species name, such as Arabidopsis thaliana found at the end of page 2 or Brassica napus in the middle of page 3, should be given in italic.

-       For most of the nutrients the acronym is given, for instance Ca, K, Mg… Why not doing the same with cadmium? You should indeed write the full name the first time you mention it but then Cd can be used instead. Same is valid to Selenium (Se).

-        

Material and Method

Section 2.1:

-       “where the concentration of cadmium (25μMol/L) was chosen according to the previous study.”  Please give the reference where this study can be found.

Section 2.2:

-       “were measured after 10 days of cadmium treatment”. In section 2.1 the authors said that “After 14 days of cadmium stress, samples were collected”. Since it is not possible to measure for instance the dry weights without collecting the plant, please clarify how the measurements were done after 10 days, if the harvest only occurred 4 days after that.

Section 2.3:  

-       The authors mention that root activity determination was performed according to the method described by Luo et al. [24]. However, first I cannot find a reference from Luo et al. in the reference list with the number 24. And the one that I find, with number 22, is a review where no methodological procedures for root activity determination are specified. Therefore please clarify and correct this, and also specify how this root activity was estimated, i.e., using a standard curve or using a formula? If this is the case, please clearly specify it.

Section 2.4:

-       Please described the formula used to determine the chlorophyl content after reading the absorbances at 649 and 665 nm.

Section 2.6:

-       Please specify how the Fv/Fm was determined. Were the leaves dark-adapted? If yes for how long.

Section 2.9:

-       Please shortly specify how the elative water content was determined.

Section 2.9:

-       The supernatant is the liquid to be measured. The supernatant was the liquid to be determined.”. Please correct it

-       Please clarify how the MDA content was calculated (formula with the molar extinction coefficient (ε) used)

Section 2.10

-       “Tissue from 0.5 g of fresh leaves was taken and 5 mL of 50 mM phosphate buffer (pH 7.8) was added. Take 0.5 g of fresh leaf tissue, ground with 5mL of 50 mM phosphate buffer (pH 7.8), and the mixture centrifugedat10,000×gfor20min.” Please correct it.

Section 2.16:

-       APX is also an enzyme from the antioxidant system, so please combine sections 2.15 and 2.16.

-       Please specify the pH of the phosphate buffer used

-       How were the SOD and POD activities determined? Please specify it!

-       Also was the APX determined using the Solarbio BCO220 Ascorbate Peroxidase Kit mentioned in section 2.15 or did the authors used the method mentioned at the end of section 2.16?! Please clarify it.

Results

-       There is no need to use two decimal places. Please change it to one.

-       “The results showed that exogenous selenium could promote the growth of cabbage seedlings under cadmium stress by increasing root water uptake and nutrient use efficiency and enhancing root activity (Figure 2I).” As far as I can see, nutrient use efficiency was not calculated, so how can the authors conclude such thing?

-       Since it is important that each figure ‘stands alone’ i.e. that the reader can understand it straight away without the need to find information elsewhere T1, T2, T3 and T4 meaning should be specified in each caption subtitle.

-       The first time that an abbreviation/acronym is given the full name should also be provided. Please use it clarify what Pn, Gs, Tr and Ci mean the first time use it (both in the text and in Figure 3 caption)

-       “indicating that exogenous selenium is antagonistic to cadmium. The cadmium content in leaves was significantly lower than that in roots after exogenous selenium treatment, indicating that exogenous selenium could inhibit the transfer of cadmium from the underground to the aboveground part by regulating the cadmium content in different parts of cabbage seedlings, thus improving the tolerance of cabbage seedlings to cadmium stress.”. This is discussion and not results section. Please move it accordingly.

References

-       Please take particular attention to this section

Only minor editing of English language required. 

Author Response

Dear Editor and the Reviewers in Horticulturae:

I am sorry to inform you that due to my checking error I submitted the wrong revised version, I am now resubmitting the final file, please refer to this manuscript, I am very sorry!

We would like to express our gratification for the opportunity to improve the paper and appreciate your guidance. We have tried our best to comply with the required revision and rectifications. This manuscript was also further edited by skilled researchers to improve the English language and grammar. We hope that concerns raised by reviewers have been well addressed. The reviewer's comments and our responses to them were listed below, and all the changes are marked in YELLOW in the revised manuscript.

We look forward to the responses from the editor and reviewers.

Thanks very much for your time and efforts to help us improve our work.

Sincerely

Kaiyue Jia

Dr. Wei Huang,

College of Horticulture, Northwest A&F University, Yangling, Shaanxi, China 712100

REVIEWER COMMENT AND AUTHORS’ RESPONSE

Thank you for all the comments from the reviewers! We addressed these comments point by point, and integrated all the responses to the revised manuscript.

Responses to comments of Reviewer 1:

Dear authors,

Congratulations on your manuscript, which I read with pleasure. The manuscript shows us an interesting set of results concerning the exogenous application of selenium to enhance cadmium stress tolerance of cabbage seedlings. Writing is in general efficient but there are problems in material and methods and results sections that need to be presented in a clearer way. Some typographical errors were also found (please see details below and carefully read the MS to find more). The discussion is very clear and supported by the results. However, there are several problems concerning the references/bibliography list, both in the way how it is presented in the text (please homogenize it), as well as concerning the numbers mentioned in the text which does not correspond to the numbers given in the reference list. The MS has great potential to be published but first these points should be corrected, and the questions below should be properly addressed before I can endorse it to publication.

Also, the MS should have line number to simplify the revision process. Please take this into consideration before submitting the improved version.

Response: Thank you very much for the professional comment.

The first time that an abbreviation/acronym is given (ROS, O2-, H2O2, MDA, SOD, POD. APX and GSH) the full name should also be provided.

Response: Thank you very much for your suggestions.

should be provided in alphabetical order.

Response: Thanks for this comment.

Species name, such as Arabidopsis thaliana found at the end of page 2 or Brassica napus in the middle of page 3, should be given in italic.

Response: Thank you very much for the professional comment.

For most of the nutrients the acronym is given, for instance Ca, K, Mg… Why not doing the same with cadmium? You should indeed write the full name the first time you mention it but then Cd can be used instead. Same is valid to Selenium (Se).

Response: Thank you for your advice.

where the concentration of cadmium (25μMol/L) was chosen according to the previous study.”  Please give the reference where this study can be found.

Response: Thank you for your comments.

1. Wang, T., Song, J., Liu, Z. et al. Melatonin alleviates cadmium toxicity by reducing nitric oxide accumulation and IRT1 expression in Chinese cabbage seedlings. Environ Sci Pollut Res28, 15394–15405 (2021). https://doi.org/10.1007/s11356-020-11689-w

2. Song A, Li Z, Zhang J, Xue G, Fan F, Liang Y. Silicon-enhanced resistance to cadmium toxicity in Brassica chinensis L. is attributed to Si-suppressed cadmium uptake and transport and Si-enhanced antioxidant defense capacity. J Hazard Mater. 2009;172(1):74-83. doi:10.1016/j.jhazmat.2009.06.143

3. Rouyi Fang Physiological regulation of serotonin and melatonin on Brassica napus growth under cadmium stress [D]. Zhejiang University,2022. DOI:10.27461/d.cnki.gzjdx.2022.002445.(In Chinese )

“were measured after 10 days of cadmium treatment”. In section 2.1 the authors said that “After 14 days of cadmium stress, samples were collected”. Since it is not possible to measure for instance the dry weights without collecting the plant, please clarify how the measurements were done after 10 days, if the harvest only occurred 4 days after that.

Response: Thank you for your professional advice and the issue has been revised.

 The authors mention that root activity determination was performed according to the method described by Luo et al. [24]. However, first I cannot find a reference from Luo et al. in the reference list with the number 24. And the one that I find, with number 22, is a review where no methodological procedures for root activity determination are specified. Therefore please clarify and correct this, and also specify how this root activity was estimated, i.e., using a standard curve or using a formula? If this is the case, please clearly specify it.

Response: Thank you very much for the professional comment.

 Please described the formula used to determine the chlorophyl content after reading the absorbances at 649 and 665 nm.

 Response: Thank you for your advice.

Please specify how the Fv/Fm was determined. Were the leaves dark-adapted? If yes for how long.

 Response: We gratefully appreciate for your valuable comment.

 Please shortly specify how the elative water content was determined.

 Response: Thank you for your professional advice and the issue has been revised.

“The supernatant is the liquid to be measured. The supernatant was the liquid to be determined.”. Please correct it

Response: Thank you very much for your suggestions.

Please clarify how the MDA content was calculated (formula with the molar extinction coefficient (ε) used)

Response: Thank you for your professional advice.

“Tissue from 0.5 g of fresh leaves was taken and 5 mL of 50 mM phosphate buffer (pH 7.8) was added. Take 0.5 g of fresh leaf tissue, ground with 5mL of 50 mM phosphate buffer (pH 7.8), and the mixture centrifugedat10,000×gfor20min.” Please correct it.

 Response: Thank you for your advice.

APX is also an enzyme from the antioxidant system, so please combine sections 2.15 and 2.16.

Response: Thanks for the comment.

Please specify the pH of the phosphate buffer used

Response: Thank you for your advice.

How were the SOD and POD activities determined? Please specify it!

Response: Thank you for your professional advice and the issue has been revised.

Also was the APX determined using the Solarbio BCO220 Ascorbate Peroxidase Kit mentioned in section 2.15 or did the authors used the method mentioned at the end of section 2.16?! Please clarify it.

 Response: Thank you for your careful reading, helpful comments.

 There is no need to use two decimal places. Please change it to one.

Response: Thank you for your advice.

 “The results showed that exogenous selenium could promote the growth of cabbage seedlings under cadmium stress by increasing root water uptake and nutrient use efficiency and enhancing root activity (Figure 2I).” As far as I can see, nutrient use efficiency was not calculated, so how can the authors conclude such thing?

Response: Thank you for your professional advice and the issue has been revised.

Since it is important that each figure ‘stands alone’ i.e. that the reader can understand it straight away without the need to find information elsewhere T1, T2, T3 and T4 meaning should be specified in each caption subtitle.

Response: Thank you for your advice.

The first time that an abbreviation/acronym is given the full name should also be provided. Please use it clarify what Pn, Gs, Tr and Ci mean the first time use it (both in the text and in Figure 3 caption)

Response: We gratefully appreciate for your valuable comment.

“indicating that exogenous selenium is antagonistic to cadmium. The cadmium content in leaves was significantly lower than that in roots after exogenous selenium treatment, indicating that exogenous selenium could inhibit the transfer of cadmium from the underground to the aboveground part by regulating the cadmium content in different parts of cabbage seedlings, thus improving the tolerance of cabbage seedlings to cadmium stress.”. This is discussion and not results section. Please move it accordingly.

  Response: Thank you for your advice.

Please take particular attention to this section

Response: Thank you very much for your suggestions.

Thank you again for your positive comments and valuable suggestions to improve the quality of our manuscript. If there are any other modifications we could make, we would like very much to modify them and we really appreciate your help.

Reviewer 2 Report

The paper ID horticulturae-2588384 entitled “Exogenous selenium enhances cadmium stress tolerance by improving physiological characteristics of cabbage (Brassica oleracea L. var. capitata) seedlings.

The manuscript is available online in preprint https://doi.org/10.20944/preprints202308.1355.v1

Although the role of selenium in crops under abiotic stress has been widely studied, the novelty of this manuscript is that it offers groups of different measurements compared to the previous one. So, I think it is could be published in horticulturae after many revisions.

However, the manuscript not well organized and difficult to follow due to no line numbering.

However, I have some observation and suggestions:

-     The abstract section is very weak, poor writing, too long with unnecessary information in this point, treatments not clear. I suggest to rewrite the abstract and shortened.

-        In the introduction; the author focused on the effect of Cd and also selenium on the nutrient’s availability and contents, however, this manuscript did not address this aspect. I suggest to review the effect of selenium and Cd on the measured variables in this study, if applicable.

-        The scientific name writing rules should be followed in the whole manuscript and references. As example page 2: Arabidopsis thaliana and page 3: Brassica napus, etc.,

-        I suggest to rewrite the last sentence in the introduction.

-        Page 4: “Seedlings were transplanted into a cavity dish”. I suppose the seeds have been planted? Not seedling.

-        Page 4: POTS? Why its capital? please follow the sentence case.

-        How the authors selected the tested selenium concentration? Also, selenium application method not clear. Is selenium applied like Cd?

-        Page 4: “the concentration of cadmium (25μMol/L) was chosen according to the previous study”. Please support this with considerable reference.

-        Lak of sufficient information regarding to the growth conditions, number of plots per plot, number of replications.

-        Please support the Se and Cd determination with references.

-        Subtitle 2.9. Not relative water content, it must be membrane stability index or electrolyte leakage.

-        Subtitle 2.10. Pro.? Abbreviation must be identified at first mentioned.

-        Too many subtitles in sampling and measurements. I suggest to group all similar variables in one subtitle, e.g., antioxidants, reactive oxygen determinations, etc.,

-        In the results; all the figures must be clearer. Please identify the abbreviations in all the figures captions.

-        In the results, the author sometimes used introductory sentences, which is the first sentence after the subtitle that not make a sense.

-        The discussion focused on the effects of cadmium on different traits and neglected the discussion of the role of selenium in raising the negative effects on cabbage. ٍI suggest deep discussion on the role of selenium on improving the different variable in the manuscript.

-        The references numbering should be revised. Some of the reference numbers in the manuscript do not match those mentioned in the Reference section, for example, in page 5 "Bajji. [27]" is number 25 in the references.

Nothing

Author Response

Dear Editor and the Reviewers in Horticulturae:

I am sorry to inform you that due to my checking error I submitted the wrong revised version, I am now resubmitting the final file, please refer to this manuscript, I am very sorry!

We would like to express our gratification for the opportunity to improve the paper and appreciate your guidance. We have tried our best to comply with the required revision and rectifications. This manuscript was also further edited by skilled researchers to improve the English language and grammar. We hope that concerns raised by reviewers have been well addressed. The reviewer's comments and our responses to them were listed below, and all the changes are marked in YELLOW in the revised manuscript.

We look forward to the responses from the editor and reviewers.

Thanks very much for your time and efforts to help us improve our work.

Sincerely

Kaiyue Jia

Dr. Wei Huang,

College of Horticulture, Northwest A&F University, Yangling, Shaanxi, China 712100

REVIEWER COMMENT AND AUTHORS’ RESPONSE

Thank you for all the comments from the reviewers! We addressed these comments point by point, and integrated all the responses to the revised manuscript.

Responses to comments of Reviewer 2:

The paper ID horticulturae-2588384 entitled “Exogenous selenium enhances cadmium stress tolerance by improving physiological characteristics of cabbage (Brassica oleracea L. var. capitata) seedlings.

The manuscript is available online in preprint https://doi.org/10.20944/preprints202308.1355.v1

Although the role of selenium in crops under abiotic stress has been widely studied, the novelty of this manuscript is that it offers groups of different measurements compared to the previous one. So, I think it is could be published in horticulturae after many revisions.

However, the manuscript not well organized and difficult to follow due to no line numbering.

However, I have some observation and suggestions:

 The abstract section is very weak, poor writing, too long with unnecessary information in this point, treatments not clear. I suggest to rewrite the abstract and shortened.

Response: Thank you very much for your suggestions.

In the introduction; the author focused on the effect of Cd and also selenium on the nutrient’s availability and contents, however, this manuscript did not address this aspect. I suggest to review the effect of selenium and Cd on the measured variables in this study, if applicable.

Response: Thank you for your professional advice.

The scientific name writing rules should be followed in the whole manuscript and references. As example page 2: Arabidopsis thaliana and page 3: Brassica napus, etc.,

Response: Thank you very much for your suggestions.

 I suggest to rewrite the last sentence in the introduction.

Response: Thanks for the comment.

Page 4: “Seedlings were transplanted into a cavity dish”. I suppose the seeds have been planted? Not seedling.

Response: Thank you very much for the professional comment.

 Page 4: POTS? Why its capital? please follow the sentence case.

Response: Thank you for your professional advice and the issue has been revised.

How the authors selected the tested selenium concentration? Also, selenium application method not clear. Is selenium applied like Cd?

Response: Thank you for your professional advice and the issue has been revised.

1.Qi, W.; Li, Q.; Chen, H.; Liu, J.; Xing, S.; Xu, M.; Yan, Z.; Song, C.; Wang, S. Selenium nanoparticles ameliorate brassica napus l. Cadmium toxicity by inhibiting the respiratory burst and scavenging reactive oxygen species. J. Hazard. Mater. 2021, 417, 125900.

2.LIU Jin-chang, XIONG Shuang-lian, MA Shuo, et al. Interactive effects of selenite and arsenite on physiological characteristics and accumulation of arsenic and selenium in rice seedlings[J]. Journal of Agro-Environment Science, 2018, 37(3): 423-430. (In Chinese)

3.Cui Xuemei,Li Binghui,Li Chunsheng et al. Mitigation effect of selenium on copper stress in rapeseed seedlings[J]. Jiangsu Agricultural Science,2018,46(04):74-78. DOI:10.15889/j.issn.1002-1302.2018.04.018. (In Chinese)

4.Sousa, G.V.; Teles, V.L.G.; Pereira, E.G.; Modolo, L.V.; Costa, L.M. Interactions between as and se upon long exposure time and effects on nutrients translocation in golden flaxseed seedlings. J. Hazard. Mater. 2021, 402, 123565.

Page 4: “the concentration of cadmium (25μMol/L) was chosen according to the previous study”. Please support this with considerable reference.

Response: Thank you for your comments.

1.Wang, T., Song, J., Liu, Z. et al. Melatonin alleviates cadmium toxicity by reducing nitric oxide accumulation and IRT1 expression in Chinese cabbage seedlings.Environ Sci Pollut Res28, 15394–15405 (2021). https://doi.org/10.1007/s11356-020-11689-w

2.Song A, Li Z, Zhang J, Xue G, Fan F, Liang Y. Silicon-enhanced resistance to cadmium toxicity in Brassica chinensis L. is attributed to Si-suppressed cadmium uptake and transport and Si-enhanced antioxidant defense capacity. J Hazard Mater. 2009;172(1):74-83. doi:10.1016/j.jhazmat.2009.06.143

3.Rouyi Fang Physiological regulation of serotonin and melatonin on Brassica napus growth under cadmium stress [D]. Zhejiang University,2022. DOI:10.27461/d.cnki.gzjdx.2022.002445.(In Chinese )

Lak of sufficient information regarding to the growth conditions, number of plots per plot, number of replications.

Response: Thank you for your professional advice.

Please support the Se and Cd determination with references.

Response: Thank you for your advice.

Subtitle 2.9. Not relative water content, it must be membrane stability index or electrolyte leakage.

Response: Thank you for your professional advice.

Subtitle 2.10. Pro.? Abbreviation must be identified at first mentioned.

Response: Thank you for your advice.

Too many subtitles in sampling and measurements. I suggest to group all similar variables in one subtitle, e.g., antioxidants, reactive oxygen determinations, etc.,

 Response: We gratefully appreciate for your valuable comment.

 In the results; all the figures must be clearer. Please identify the abbreviations in all the figures captions.

Response: Thank you very much for the professional comment.

In the results, the author sometimes used introductory sentences, which is the first sentence after the subtitle that not make a sense.

Response: Thank you very much for your suggestions.

The discussion focused on the effects of cadmium on different traits and neglected the discussion of the role of selenium in raising the negative effects on cabbage. ٍI suggest deep discussion on the role of selenium on improving the different variable in the manuscript.

Response: Thank you for your comments.

The references numbering should be revised. Some of the reference numbers in the manuscript do not match those mentioned in the Reference section, for example, in page 5 "Bajji. [27]" is number 25 in the references.

Response: Thank you very much for your suggestions.

Thank you again for your positive comments and valuable suggestions to improve the quality of our manuscript. If there are any other modifications we could make, we would like very much to modify them and we really appreciate your help.

Round 2

Reviewer 2 Report

I think that the authors made several modifications and improvements making the manuscript better.

However, in other cases some highlighted sentences not really changed compared to the first version particularly in Abstract and Introduction.

The Abstract still a delema (without fundamental change, and too long; 390 word). I suggest to remove the sentence started from "cabbage" in Line 12-15.

Line 101-102: Please insert the references you relied on here.

The clarity of the figures is not the best.

No

Author Response

Dear Editor and the Reviewers in Horticulturae:

We would like to express our gratification for the opportunity to improve the paper and appreciate your guidance. We have tried our best to comply with the required revision and rectifications. This manuscript was also further edited by skilled researchers to improve the English language and grammar. We hope that concerns raised by reviewers have been well addressed. The reviewer's comments and our responses to them were listed below, and all the changes are marked in RED in the revised manuscript.

We look forward to the responses from the editor and reviewers.

Thanks very much for your time and efforts to help us improve our work.

Sincerely

Kaiyue Jia

Dr. Wei Huang,

College of Horticulture, Northwest A&F University, Yangling, Shaanxi, China 712100

REVIEWER COMMENT AND AUTHORS’ RESPONSE

Thank you for all the comments from the reviewers! We addressed these comments point by point, and integrated all the responses to the revised manuscript.

Responses to comments of Reviewer :

I think that the authors made several modifications and improvements making the manuscript better.

However, in other cases some highlighted sentences not really changed compared to the first version particularly in Abstract and Introduction.

Response: Thank you very much for the professional comment.

The Abstract still a delema (without fundamental change, and too long; 390 word). I suggest to remove the sentence started from "cabbage" in Line 12-15.

Response: Thank you for your professional advice and the issue has been revised.

Line 101-102: Please insert the references you relied on here.

Response: Thank you very much for your suggestions.

The clarity of the figures is not the best.

 Response: Thank you for your professional advice and the issue has been revised. We have also included figures from the original results for your reference.

Thank you again for your positive comments and valuable suggestions to improve the quality of our manuscript. If there are any other modifications we could make, we would like very much to modify them and we really appreciate your help.
